# A step towards measuring connectivity in the deep sea: elemental fingerprints of mollusk larval shells discriminate hydrothermal vent sites

Vincent Mouchi[1], Christophe Pecheyran[2], Fanny Claverie[2], Cécile Cathalot[3], Marjolaine Matabos[4], Yoan Germain[3], Olivier Rouxel[3], Didier Jollivet[1], Thomas Broquet[1], Thierry Comtet[1]

[1]Sorbonne Universite, CNRS, Adaptation et Diversité en Milieu Marin, AD2M, Station Biologique de Roscoff, F-29680, Roscoff, France
[2]Universite de Pau et des Pays de l'Adour, E2S UPPA, CNRS, IPREM, Avenue de l'Université, BP 576 64012 PAU cedex, France
[3]Geo-Ocean, Univ Brest, CNRS, IFREMER, UMR6539, F-29280 Plouzané, France
[4]IFREMER REM-EEP, Technopôle Brest Plouzané, 29280 Plouzané, France

*Correspondence to*: Vincent Mouchi (vmouchi@gmail.com)

**Abstract.** Deep-sea hydrothermal-vent systems are under investigation for base and precious metal exploitations. The impact of mining will depend critically on the ability of larval dispersal to connect and replenish endemic populations. However, assessing connectivity is extremely challenging, especially in the deep sea. Here, we investigate the potential of elemental fingerprinting of mollusc larval shells to discriminate larval origins between multiple hydrothermal sites in the Southwest Pacific Ocean. The gastropod *Shinkailepas tollmanni* represents a suitable candidate as it uses capsules to hold larvae before dispersal, which facilitates sampling and ensures mineralization occurs on the site of origin. Multielemental microchemistry was performed using cutting-edge femtosecond laser ablation Inductively Coupled Plasma Mass Spectrometry analysis to obtain individual measurements on 600 encapsulated larval shells. We used classification methods to discriminate the origin of individuals from 14 hydrothermal sites spanning over 3,500 km, with an overall success rate of 70%. When considering less sites within more restricted areas, reflecting dispersal distances reported by genetic and modelling approaches, the success rate increased up to 86%. We conclude that individual larval shells register site-specific elemental signatures that can be used to assess their origin. These results open new perspectives to get direct estimates on population connectivity from the geochemistry of pre-dispersal shell of recently settled juveniles.

## 1 Introduction

Deep-sea hydrothermal activity produces massive polymetallic sulphide deposits from the ascent and precipitation of hydrothermal fluids enriched in metals originating from the oceanic crust (Fouquet et al., 1991; Binns and Scott, 1993; Humphris et al., 1995; Hannington et al., 2011). Vent sites are economically interesting to mining companies (Hoagland et al., 2010; Petersen et al., 2016; Radziejewska et al., 2022), which initiatives are currently under evaluation worldwide (Boschen

et al., 2013; Dunn et al., 2018). Deep-sea mining is expected to strongly disturb hydrothermal-vent ecosystems (Van Dover, 2014; Danovaro et al., 2017, 2020), yet very little is known on the resilience of their associated communities (Dunn et al., 2018; Suzuki et al., 2018).

Hydrothermal-vent fields are ephemeral and patchy habitats, separated by hundreds of meters to thousands of kilometres. The resilience of vent populations therefore depends critically on their level of connectivity, i.e., the migration of individuals between vent sites. As a basic process of population dynamics, connectivity is key to understand the persistence of vent communities as well as the potential of vent species to colonize new areas and habitats (e.g., Adams et al., 2012; Levin et al., 2016). As such, it is of major interest for biodiversity conservation and the design of deep-sea marine protected areas (Combes

et al., 2021), for example in the framework of the 30x30 initiative, which aims to protect at least 30% of the world's oceans by 2030 (O'Leary et al. 2019). In the particular context of deep-sea mining, connectivity has been identified as a key scientific knowledge gap (Gollner et al., 2017; Miller et al., 2018; Smith et al., 2020; Amon et al., 2022) that should be tackled taking into account its interactions with climate change (Levin et al., 2020). For vent benthic species, pelagic larvae are the major, if not the only vector of dispersal between distant and well-separated populations living at active sites, and a means to colonize

new territories. Accurately estimating larval dispersal is therefore critical to understand the demographic trajectory of hydrothermal metapopulations, especially in the context of habitat destruction (Amon et al., 2022). It is however extremely difficult to catch minute-sized larvae released at great depths (although some successful attempts were reported: Comtet et al., 2000; Mullineaux et al., 2005), and almost impossible to directly track their journey between their initial release and final settlement locations.

Indirect methods exist to assess connectivity, but have some limitations. Larval dispersal modelling aims at estimating the spatial extent of larval transport in a given area (Mitarai et al., 2016; Vic et al., 2018; Yearsley et al., 2020). It uses physical oceanographic data and species-specific biological features, such as the planktonic larval duration and larval behaviour (e.g., vertical migration), but biological and hydrodynamic parameters are hardly constrained, inducing uncertainties in the larval travelling distance (McVeigh et al., 2017). Other indirect methods based on genetic data (Baco et al., 2016; Breusing et al.,

2016; Tran Lu Y et al., 2022) are often inconclusive in broadcast-spawning species because large and very fertile populations experience weak genetic drift, resulting in low population differences regardless of migration rates (Gagnaire et al., 2015). The genetic approach can use more direct methods, e.g., based on parentage analyses, but these are applicable only when a significant fraction of individuals is sampled (Gagnaire et al., 2015), which is unworkable in the deep sea.

Although used for many years to assess directly larval dispersal in coastal environments (Levin, 1990; Levin et al., 1993;

Zacherl et al., 2003; Becker et al., 2005, 2007; Carson, 2010; Fodrie et al., 2011; Kroll et al., 2016; Honig et al., 2020; Bounket et al., 2021), elemental fingerprinting of biogenic carbonate structures (such as otoliths or shells) has never been applied to vent species (Mullineaux et al., 2018; Cunha et al., 2020), even though its interest for deep-sea environments has been suggested over 15 years ago (Levin, 2006). This approach relies on the record in calcified structures of the chemical composition of the water in which they have formed (Thorrold et al., 2007). When spatial differences occur in seawater

composition, these can be imprinted in the biominerals of organisms living there, with potential modifications due to

environmental conditions (e.g., temperature: Thorrold et al., 2007) and animal metabolism (Mouchi et al., 2020). In fish and molluscs, calcified structures produced during the early larval stage and preserved thereafter can be used to infer their natal geographic origin, provided that they have been built in chemically contrasting sites (Becker et al., 2007; Thorrold et al., 2007). This approach sounds promising in hydrothermal systems for which different chemical signatures are expected at various spatial scales (Hannington et al., 1991; Le Bris et al., 2003; Toffolo et al., 2020). Besides, newly developed cutting-edge analytical methods now allow precise geochemical measurements on individual 100 µm-size larval shells. Assigning the origin of juveniles from elemental fingerprints will provide rare evidence of single-generation connectivity ranges and identify which sites correspond to sources and sinks for populations over a wide area. This type of information can subsequently be used to identify sites of higher priority for preservation and protection from disturbance to ensure the resilience of populations (e.g., no-mining areas known as Areas of Particular Environmental Interest, or APEIs, International Seabed Authority, 2011; Dunn et al. 2018; Preservation Reference Zones and Impact Reference Zones; International Seabed Authority, 2019). Moreover, identifying source and sink population and migration rates for many species would therefore help identify the species vulnerable to disturbance or even destruction of a specific habitat.

The first step towards using elemental fingerprinting to estimate connectivity in the deep sea is to determine whether the elemental composition of larval shells records a distinct chemical signature between hydrothermal vent populations. Here, we define the elemental fingerprints of shells of encapsulated larvae of the gastropod *Shinkailepas tollmanni* (L. Beck, 1992) at multiple hydrothermal sites over 3,500 km in the Southwest Pacific (Fig. 1). *Shinkailepas tollmanni* (Fig. 2) is an abundant vent limpet which lays eggs in capsules deposited on the shell of the larger symbiotic gastropod *Ifremeria nautilei* (Bouchet and Warén, 1991) living in diffuse venting areas. Within these capsules, eggs develop into veliger larvae whose carbonate shells incorporate elements from the surrounding habitat water before dispersing. The challenge is to get high-quality geochemical measurements from the 100 µm-size encapsulated larval shells (Fig. 2f) to highlight the potential of this approach to discriminate sites of origin. To explore the use of elemental fingerprints of larval shells in this species, several questions were investigated. Are there enough chemical contrasts between hydrothermal vent sites for elemental fingerprinting? Do elemental fingerprints of larval shells reflect their natal place? What are the main elements responsible for the elemental fingerprint of larval shells? At which spatial scale is elemental fingerprint more accurate? Does the elemental fingerprint of larval shells correspond to that of habitat water, in the sense that the habitat water fingerprint could be used as a reference for the shell fingerprint?

## 2 Materials and Methods

### 2.1 *Shinkailepas tollmanni*

*Shinkailepas tollmanni* (L. Beck, 1992), first described from the Manus basin (Beck, 1992), is an abundant phenacolepadid gastropod widely distributed in the Western Pacific back-arc basins, covering depths ranging from 1300 to 3300 m (Sasaki et al., 2010; Collins et al., 2012; Boulart et al., 2022; Poitrimol, 2022). Adults live in the three faunal assemblages common to

these areas, *i.e.* the *Bathymodiolus* spp., *Ifremeria nautilei* and *Alviniconcha* spp. communities (Podowski et al., 2009; Collins et al., 2012; Poitrimol, 2022), with higher abundances on shells of *I. nautilei* (Poitrimol, 2022). As early hypothesized by Beck (1992), *S. tollmanni* likely grazes on free-living bacteria or organic matter from vent surfaces and mollusc shells it lives on (Van Audenhaege et al., 2019; Suh et al., 2022). *S. tollmanni* is gonochoric with continuous gametogenesis (Poitrimol et al., *in press*) and fertilization is internal (Sasaki et al., 2010). Females lay egg capsules on shells of living *Ifremeria nautilei* as first hypothesized by Beck (1992), up to the veliger stage 170-180 µm length (Yahagi et al., 2020). Whether the relationship between *S. tollmanni* and *I. nautilei* for egg-laying is obligate is unknown: to our knowledge the presence of egg capsules of *S. tollmanni* on *Bathymodiolus* or *Alviniconcha* shells, or on vent surfaces, has never been reported, but Yahagi et al. (2020) reported egg capsules on conspecific adult shells. Based on morphological characteristics (*e.g.* size of encapsulated veligers *vs* size of postlarvae), larvae of *S. tollmanni* are considered planktotrophic, and are assumed to disperse for up to one year (Yahagi et al., 2020). Although no direct evidence exists, larvae of *S. tollmanni* are assumed to disperse in surface waters as shown for other Phenacolepadidae (Yahagi et al., 2020). Such dispersal abilities could explain the genetic homogeneity observed among populations of *S. tollmanni* at the scale of the study area (Yahagi et al., 2020; Poitrimol et al., 2022). Recruitment is likely continuous although episodic larval supply could also occur (Poitrimol et al., *in press*).

## 2.2 Biological sampling

Egg capsules of *Shinkailepas tollmanni* were sampled from the apex and spire crests of the shell of large symbiotic gastropods *Ifremeria nautilei* that were collected during the Chubacarc oceanographic cruise (Hourdez and Jollivet, 2019) aboard the research vessel *L'Atalante* in spring 2019. Site coordinates are indicated in Supplementary Information S1. Individuals of *I. nautilei* were collected using the hydraulic claw of the remotely operated vehicle (ROV) Victor 6000 and brought back to the surface in an insulated basket (this corresponds to one biobox or sampling replicate). On board, their shells were examined for egg capsules containing living shelled embryos of *S. tollmanni* using a dissecting microscope. To this extent, egg capsules were maintained immersed using the local deep-sea water contained in bioboxes before opening, and larvae that had reached the veliger stage (swimming larvae with visible shell) were recovered with a Pasteur pipette. The presence of a calcified shell was confirmed after examining the collected larvae under polarized light. All the plastic material was washed with nitric acid and rinsed with pure water before use. After collection, the larvae were stored dry at -20°C in groups of 10 to 100 in 2-mL microtubes until processed according to the protocol described below.

## 2.3 Habitat water composition analysis

Habitat water surrounding the *Ifremeria* communities was sampled in one point using the In-Situ Fluid Sampler (PIF) manipulated by the ROV arm at each surveyed vent site during the cruise prior to animal collection. Note that not all sites investigated for water composition in the *Ifremeria* communities comprised *S. tollmanni* individuals. As described in previous studies (Cotte et al., 2015), such in situ samplers allow to recover low-temperature fluids with minimal metal contamination due to the use of metal-free materials, and perform if needed in situ filtration. Here, habitat water samples were filtered during

sampling by mounting on-line Acrodisc® Syringe Filters incorporating a 0.22 µm Supor® hydrophilic polyethersulfone membrane. Samples are then acidified to pH 1.8 with ultrapure HCl and stored in LDPE bottles. Compositions of both major and minor elements were measured by high-resolution inductively coupled plasma mass spectrometry (HR-ICPMS) Element XR operated at Ifremer following previously described methods (Rouxel et al., 2018; Konn et al., 2022). In short, isotopes $^{44}Ca$, $^{39}K$, $^{24}Mg$, $^{32}S$, $^{28}Si$, $^{56}Fe$, $^{55}Mn$, $^{63}Cu$, $^{66}Zn$, $^{27}Al$, $^{51}V$, $^{31}P$ are measured in medium mass resolution mode, while isotopes

$^{7}Li$, $^{11}B$, $^{85}Rb$, $^{88}Sr$, $^{89}Y$, $^{98}Mo$, $^{238}U$, are measured in low mass resolution mode. Analyses were done on aliquots, diluted 100-fold with 0.28 M $HNO_3$ containing an internal spike of In (at 2 ng.g$^{-1}$ each). Solutions were introduced into the plasma torch using a quartz spray chamber system equipped with a microconcentric PFA nebulizer operating at a flow rate of about 100 µL.min$^{-1}$. Precision and accuracy were determined for each analytical run by repeat analysis of internal reference material of similar range in composition as the samples (trace metal dopped seawater) also spiked with In (2 ng.g$^{-1}$). For each element,

ICPMS sensitivity was calibrated using a set of matrix-matched in-house standard solutions corresponding to seawater matrices and IAPSO Standard Seawater. Precision was better than 5% (2 s.d.) for reported elements. Detection limit was determined as 3 standard deviation of repeated reagent blank's signal processed through the same protocol as for unknown samples.

## 2.4 Larval shell preparation

     Details of the shell preparation protocol were described in (Mouchi et al., 2023). Briefly, all plastic materials handled during

the cleaning and preparation of the samples were acid-cleaned using 10% PrimarPlus-Trace analysis grade $HNO_3$ (Fisher Chemicals, 10098862) and rinsed with ultrapure water in Teflon beakers, in a clean lab under a laminar flow cabinet (ISO5). No metallic objects were used in contact of the samples at any stage of the sample cleaning. A protocol modified from (Becker et al., 2005) was used to remove the soft body from the shells. A solution of Optima-grade $H_2O_2$ at 30% (Merck KGaA, 107298) was buffered with 0.05 mol.L$^{-1}$ NaOH (Suprapur) to obtain a pH of 8.5. The final $H_2O_2$ concentration was

approximately 15%. This solution was used to digest the larval tissues in a glass container overnight. The resulting cleaned shells were then rinsed with ultrapure water and collected by a sable brush to place them on an Extra Pure carbon adhesive tabs (Science Services) on a microscope slide. After the remaining ultrapure water dried out on the tabs, the slides were placed in a clean airtight plastic box until analysis.

## 2.5 Individual larval shell elemental analysis

The elemental composition of cleaned larval shells was measured at the Institut des Sciences Analytiques et de Physico-Chimie pour l'Environnement et les Matériaux (Université de Pau et des Pays de l'Adour), with an Agilent 8900 ICP-MS Triple Quad coupled with a femtosecond laser ablation system (Novalase SA). The femtosecond (360fs) laser ablation system was set to generate pulses of 23 µJ at 50 Hz. It is equipped with a 2D galvanometric scanner which allows moving the laser beam at the surface of the sample at high speed. Each shell was ablated following the trajectory of a scanned disc of 100 µm diameter (with

7.5 µm step) at 1 mm.s-1 speed rate. For each acquisition, the disc was performed twice to ensure the total ablation of the larval shell, while preventing excessive ablation of the tape. Some zones of the tape without sample were also ablated separately

in the same conditions in order to check for potential contamination. No evidence of critical signal was observed on this tape for the elements of interest. Helium, at a 450 mL.min$^{-1}$ flow rate, was chosen to transport the ablated particles to the ICPMS. The ICPMS was operated in MS/MS mode with 10 mL.min$^{-1}$ of H$_2$. Measured elements were $^{24}$Mg, $^{52}$Cr, $^{55}$Mn, $^{56}$Fe, $^{63}$Cu, $^{66}$Zn, $^{75}$As, $^{88}$Sr, $^{114}$Cd, $^{120}$Sn, $^{121}$Sb, $^{138}$Ba, $^{208}$Pb and $^{43}$Ca as the internal standard to control the ablated volume. Dwell time was 0.005 second for each element. Each sample representing an extremely small amount of material, in the order of micrograms, the complete ablation of the sample took place in only a few seconds (Supplementary Information S2). Calibration was performed by the successive measurements of the reference glass materials NIST SRM 610, 612 and 614 at the beginning, the middle and the end of each analytical session using $^{43}$Ca as internal standard. Data reduction was performed using an in-lab developed software FOCAL 2.41. Accuracy of the measurements was checked by measuring the otolith certified reference materials (CRM) FEBS-1 and NIES-22 against the preferred values from the GeoRem database (Jochum et al., 2005), when available.

## 2.6 Data processing

All data were processed using the Matlab software (Mathworks, www.mathworks.com, v. R2022a). Correlation between habitat water and larval shell compositions was assessed using the Pearson determination coefficient $r^2$ and the Spearman correlation coefficient $rho$. Principal Component Analysis (PCA) was performed to explore the variance of habitat water and larval shell compositions according to hydrothermal sites. For PCA, habitat water and larval shell data were transformed to the cubic root (Chen and Deo, 2004) to improve data dispersion and interpretation, and normalized. Larval shell measurements were only transformed to the cubic root for classifications. Classification models (for site and area discrimination based on larval shell geochemistry; Supplementary Information S3) were performed using the Matlab *ClassificationLearner* application and five equal folds (i.e., disjoint divisions) of the dataset for cross-validation: each fold was successively used as validation fold to assess the model based on the remaining four folds, and a mean classification rate was calculated from the five validation folds. Although linear discriminant analysis has been widely used as the preferred classification method for assessing connectivity from elemental fingerprints in coastal environments (Becker et al., 2005; Gomes et al., 2016), alternative, more accurate and successful methods have been proposed (Mercier et al., 2011; Dixon and Brereton, 2009). We here applied 31 classification methods (available in *ClassificationLearner*) on the chemistry of encapsulated larval shells in order to identify the best-fitting model (obtained from one classification method and a specific list of predictors) to determine their geographical origin. For each dataset (all sites, sites in Manus and Woodlark, sites in East region and Woodlark, and areas in East region and Woodlark), we successively performed all models with different numbers and combinations of elements. Model significance was assessed by comparing its classification success rate with that of randomized data (1000 runs) using a modified version of the Matlab code (Supplementary Information S4) by (White and Ruttenberg, 2007), which was initially restricted to the discriminant analysis.

## 3 Results and Discussion

### 3.1 Are there enough chemical contrasts between hydrothermal vent sites for elemental fingerprinting?

*Shinkailepas tollmanni* egg capsules are in contact with the mixture between seawater and hydrothermal fluid associated with the habitat of *Ifremeria nautilei*, in proportions that vary depending on the vent activity, local currents and turbulence (Podowski et al., 2010). This mixture will hereafter be referred to as the 'habitat water'. Because habitat water likely influences the composition of biogenic carbonates, we first investigated the potential differences in the elemental composition of the habitat water surrounding *Ifremeria* communities, which may be defined by the geological nature of the underlying substrate

and by phase separation processes (Hannington et al., 2013) leading to the extensive precipitation of metal sulphide and oxide minerals. Distinct water compositions will be an indication that larval shells formed in the capsules may record potentially useful differences in their elemental composition.

Habitat water elemental composition differed among only some of the sampled sites (Fig. 3a-b and Supplementary Information S5), as shown by a Principal Component Analysis (PCA) conducted on 40 samples with 19 elements. Amongst the most

differentiated sites, the Mangatolo Triple Junction is well separated from the other sites (Fig. 3a), together with two nearby chimneys at the sites Aster'X and Stéphanie sampled in the field of Fatu Kapa (Futuna). Habitat water compositions were however overlapping for the other sites despite the great geographic distances separating them, especially in the western region (Manus and Woodlark basins; Fig. 3a). These results indicated that the chemistry of the habitat water was variable enough to distinguish some sites, depending on the geographic scale considered. The next question was thus to assess to what extent

encapsulated shells record these slight differences, and at what scale.

We therefore analysed a total of 600 encapsulated larval shells of *S. tollmanni* collected from 14 vent sites in the Southwest Pacific. The femtosecond laser ablation system coupled ICP-MS/MS (8900 in semiconductor configuration) allowed us to measure 13 elements (Ca excluded) from each encapsulated larval shell despite their minute size (100 µm empty spheres, approx. 2 µm thickness; Fig. 2e). Measured abundances (Supplementary Information S1) ranged from 0.5 ng.g$^{-1}$ (for Sb) to

10.2 103 µg.g$^{-1}$ (for Mg).

Considering the abundance variations of each element (Supplementary Information S6), no single element or pair of elements is discriminant, although some substantial differences exist, enabling particular sites to be identified. This was particularly visible for Kulo Lasi (a volcanic caldera) where larval shells are strongly depleted in Mn. Some elements, namely Mg, Fe, Zn, Ba, and Pb, displayed a clear site-to-site variation, but values systematically overlapped at several sites, even across distant

areas. Other elements had low geographic power. The abundance homogeneity of Cr, As, Cd and Sb across sites was probably due to their extremely low values in shells (generally in the range 1-10 µg.g$^{-1}$; Supplementary Information S1). Copper displayed slightly higher and more variable abundances, between 0.7 and 643.5 µg.g$^{-1}$ (Supplementary Information S1).

Contrary to the habitat water data presented above, a simple PCA from the larval shell composition showed no obvious geographic clustering (Fig. 3c-d). Still, some geographic signal was visible on PC2, which appeared to reflect Mn and Ba in

particular. Shells from Kulo Lasi and Tu'i Malila generally exhibit lower abundances of these elements compared to those of

the other sites. Thus, in a second step we turned to more powerful, discriminant approaches to assess whether there is enough signal in the data to assign the origin of individual shells. That approach was possible due to the significant number of measurements from each site, which was not the case for habitat waters.

### 3.2 Do elemental fingerprints of larval shells reflect their natal place?

Identification of the site of origin of individual larvae was possible with a 70.0% mean accuracy (Fig. 4), based on classification methods using Mg, Mn, Fe, Zn, Sr, Ba, and Pb as predictors (Supplementary Information S3). For comparison, the success rate by random assignment was 20.1% using randomized origins (p < 0.001). This success rate appeared to be similar to or better than those obtained from models of coastal environments at such a scale (Becker et al., 2005; Simmonds et al., 2014; Gomes et al., 2016). With this model (i.e., results of a classification, represented by a confusion matrix as illustrated on Fig.

4), correct individual assignment ranged from 25.0% for La Scala (Woodlark) to 93.5% for Kulo Lasi (Futuna). Looking at these two extremes, the weak assignment success to La Scala may be due to the small number of specimens sampled at this location (n=4) whereas the high assignment success to Kulo Lasi is mainly due to the low abundance of Mn in larval shells (Supplementary Information S6), in compliance with the very low concentration of this element at this site (Supplementary Information S5). Phoenix (North Fiji) displayed the most heterogeneous larval shell composition, as illustrated by the wide

distribution of values for each element in shells from this location compared to the others (see boxplots in Supplementary Information S6). As a consequence, numerous larvae from other sites were wrongly assigned to this location, and a third of larvae originating from this site were misassigned to other locations by the model (69.8% accuracy for Phoenix). Caution must therefore be taken when assigning a larva to Phoenix.

However, a significant improvement of the classification was obtained when focusing on smaller geographic scales, which is

justified when considering additional information on population connectivity based on other methods, such as larval transport modelling or population genetics. In particular, simulations of larval transport have suggested that Manus and the eastern (Lau/Fiji/Futuna) basins are not directly connected by larval dispersal, and that Woodlark acts as a sink area for both regions (Mitarai et al., 2016). Population genetics reported partially congruent conclusions: migration was found to be strongly limited between these two regions for several gastropod species with a pelagic larval phase (Tran Lu Y et al., 2022; Plouviez et al.,

2019; Breusing et al., 2023), although Woodlark could act as a 'stepping stone' allowing limited connectivity (Boulart et al., 2022; Poitrimol et al., 2022). For *S. tollmanni*, although no population differentiation has been observed between these regions based on the mitochondrial genome (Poitrimol et al., 2022; Yahagi et al., 2020), genome-wide data obtained from RAD-sequencing recently identified a strong genetic break indicating a lack of dispersal between Manus and the eastern region (Tran Lu Y, 2022). The Woodlark ridge was however a recipient for migrants from the two regions, representing a tension zone

(Tran Lu Y, 2022, pp. 87-89). Based on this information, we split our data in two datasets, comprising sites of the western region (Manus/Woodlark) on the one hand, and eastern region plus Woodlark on the other hand (considering that a larva settling in the eastern region could originate from Woodlark but not from Manus). Within the western region, the overall accuracy reached 86.5%, using Mn and Ba as predictors (Fig. 5a), compared to 26.1% by random assignment (p<0.001). Site

accuracy ranged from 54.5% for North Su to 100% for Solwara 8. Similarly, the best fitting model for the eastern sites (including La Scala in the Woodlark Basin) showed an overall accuracy of 77.1% with Mg, Mn, Fe, Zn, Sr, Ba and Pb as predictors (Fig. 5b), compared to 23.7% by chance (p<0.001). Another model using the same dataset, method and predictors, but focusing on areas instead of sites (Fig. 5c), obtained a slightly lower accuracy (75.3% vs 35.3% by chance; p<0.001). The site model presents better accuracy due to the strong difference in Mn abundance between the sites from Futuna (Kulo Lasi and Fatu Kapa) that are considered as one single class by the area model.

In the future, if additional alternative information points to dispersal restricted within a single area in the eastern region, the assignment success to the site of origin for each area would reach 97.7%, 98.6%, and 100% for Lau Basin, Mangatolo, and Futuna volcanic arc, respectively (Fig. 6), while random assignment would be correct for 53.7% (p<0.001), 66.8% (p<0.001), and 75.4% (p<0.001) of the larvae, respectively. This would be particularly interesting as genomic data failed to provide good individual assignments at such small spatial scales because of both the size of the populations and their level of exchange (Tran Lu Y et al., 2022).

Selecting specific elements for classification appears to be a pre-requisite for individual assignments, as it was previously suggested (Mercier et al., 2011); all predictors had a different impact on the determination of origin. Some had a negative impact, and especially those with low values that tended to induce noise rather than a discriminatory signal, as their accuracy is probably reduced. This was the case here for Cr, As, Cd, Sb, and to a lesser extent, Cu and Sn. Removing these elements systematically improved the validation of geographic models. Moreover, our selected predictors differed depending on the spatial scale under scrutiny. Indeed, tracking back the origin of a specimen from the eastern region to the Lau Basin can be performed with 7 predictors (Fig. 5c), whereas only two are needed to identify the site of origin within the Lau Basin with a much better success rate (Fig. 6a), under the hypothesis that the individual originates from that area.

In most cases, the Support Vector Machines (SVM) method appeared to generate the best classification model (Supplementary Information S3). Dixon and Brereton (2009) explored different model types from various datasets and concluded that SVM generally performs better than other methods when data normality is not met, which is the case here. Linear discriminant analysis, widely used in the literature, was selected as the most appropriate method for only one of the seven models presented in this work, with a well-defined clustering dataset (Fig. 6a; see Supplementary Information S3).

We aimed to check whether there is, indeed, a site-specific fingerprint. Our dataset of larval shells therefore included at least two different sampling replicates from *Ifremeria* communities from one site for more than half of the sampled locations. This sampling strategy clearly showed that shell compositions were more variable between site than within sites (Supplementary Information S7), which strengthened the fact that most vent sites have their own elemental signatures.

There are, however, some limitations. First, the temporal variations of the elemental composition of the habitat water (linked to the vent activity) are unknown, and would require temporal monitoring, with series of samples over several generations of larvae. This would allow to assess the stability of the reference elemental fingerprints, and thus minimise the misassignment risk associated with the use of elemental fingerprints on individuals collected at different periods. A relatively stable vent activity has been monitored over seven years within a vent mussel assemblage at the Eiffel Tower edifice (Lucky Strike vent

field, Mid-Atlantic Ridge; Van Audenhaege et al., 2022), although data on back-arc basin context are not currently available. This can be of particular interest for species such as *S. tollmanni*, which exhibits continuous reproduction (Poitrimol, 2022),

Poitrimol et al., *in press*): the elemental fingerprinting for such hydrothermal contexts should be stable at the scale of several consecutive cohorts of larvae. Second, our models are restricted to the sites we sampled, and other hydrothermal sites may still have to be discovered at the scale of a given area. Our sampling is however substantial and represents the largest effort done so far in deep-sea vents, with 14 hydrothermal sites, including two newly discovered vent fields (La Scala, in the Woodlark Basin: Boulart et al., 2022; and Mangatolo). To circumvent such a limitation, which is not specific to deep-sea settings but is

also found in coastal systems, Simmonds et al. (2014) proposed to interpolate elemental fingerprints using a kriging method in locations where no samples are available. However, if this method is of interest in coastal environments where shell fingerprints could rely on the distance from specific sources (e.g., rivers), it is not relevant here. Indeed, hydrothermal vent systems are patchy environments, separated by hundreds to thousands of kilometres of background seawater, and the composition of these vents are often specific and strongly dependent on the subsurface phase separation processes (Charlou et

al., 2000) that cannot be interpolated. Still, our models should at least help provide the area of origin, even if the exact site is unknown. Additional samples collected from other sites are therefore needed to complement the models for future assignment of post-dispersal stage individuals.

### 3.3 Can habitat water composition be used as reference for larval shell elemental fingerprint?

As suspected, element concentrations in habitat water appeared to be poor predictors of their concentrations in shells of

encapsulated larvae. The comparison of the element concentrations in habitat water and in shells of encapsulated larvae was carried out for six elements (Mg, Mn, Fe, Cu, Zn, and Sr) measured in both the water and the shells for 13 sites. Using elemental ratios to Ca, commonly reported for carbonates (e.g., Thorrold et al., 2007), a weak but significant positive correlation between water and shells was observed only for Cu/Ca (Pearson correlation) and Fe/Ca (Pearson and Spearman correlation) (Table 1), which suggested that element concentrations in habitat water are poor predictors of their concentrations in shells of

encapsulated larvae. Because hydrothermal vent fluids are released in pulses and not continuously, bias might occur during fluid sampling. To avoid such bias, we used Mn as a proxy for the hydrothermal fluid dilution in seawater (as this element is mainly introduced in seawater from hydrothermal activity), and compared the elemental ratios to Mn between habitat water and larval shells. A weak but significant positive correlation was noted for Mg/Mn, Cu/Mn, Zn/Mn and Sr/Mn ($r^2 = 0.43\text{-}0.64$, p-value $< 0.001$; Table 2), while Fe/Mn was not significantly correlated. This showed that the elemental composition of the

vent fluid influences the elemental fingerprint in the larval shells, at least for some elements.
Discrepancies between fluid and carbonate elemental composition have already been observed, including in molluscan larvae (Strasser et al., 2008a), and is likely due to vital effects (Weiner and Dove, 2003; Ulrich et al., 2021). Several factors can be at play to cause this difference. Firstly, *S. tollmanni* larval shells are formed in capsules, laid by the females, which contain an intracapsular fluid of unknown composition, particularly regarding the elemental composition. In particular, as shown above,

Fe/Mn was not significantly correlated between shells and water, which might be explained by a fraction of fluid Fe being

trapped in the formation of Fe-oxides that are not present in the *S. tollmanni* capsules. These capsules also represent a barrier against the surrounding environment, with an unknown permeability to elements, which may change during larval development, as shown in the coastal gastropod *Crepidula fornicata* (Maeda-Martínez, 2008). Secondly, metabolic activity requires specific elements (Mg, Ca, Fe, Cu…) sampled from the environment to operate (Abreu and Cabelli, 2010; Vest et al.,

2013), which may hamper their incorporation in the shell. Thirdly, larvae may control the chemistry of the fluid located between their body and the calcifying carbonate (i.e., where the shell is built) as shown in mussel larvae (Ramesh et al., 2017). During this process, they are able to change the pH of the fluid to increase calcification rate (Ramesh et al., 2017), which in turn favours Ca substitutions with other metals (Watson, 2004; DePaolo, 2011). Lastly, mineralogy is also responsible for the preferred uptake of some elements over others in the carbonate lattice because of their atomic radii and charges. For instance,

Sr displays a significantly higher (several orders of magnitude) incorporation coefficient in aragonite (which is the calcium carbonate mineralogy of mollusc larval shells; Weiss et al., 2002) compared to calcite due to their spatial structure (Littlewood et al., 2017). Following this line, our data clearly evidenced that, although influencing the elemental composition of larval shells, the elemental concentrations of habitat water are not reliable direct predictors of shell chemistry, and thus habitat water cannot be used as a reference to assign the origin of individuals, as reported in fish otoliths and larval molluscan shells (Kroll

et al., 2016; Strasser et al., 2008b; Bouchoucha et al., 2018). Therefore, references of habitat water composition cannot reliably replace references from larval shells for identification of natal origin of migrants. Only measurements from larval shells, such as the dataset we provide here, can be used as references for the identification of natal origin of migrants.

## 4 Conclusions

The impact of upcoming deep-sea mining programs on hydrothermal-vent communities throughout the world will strongly

depend on the connectivity of populations. Our study assessed the accuracy of elemental fingerprinting of deep-sea hydrothermal mollusc larval shells, which is the first step to determine the geographic origin of individuals at different spatial scales. Sampled hydrothermal vent fields were characterized by distinct chemical compositions that are partially reflected in shells of encapsulated *S. tollmanni* larvae.

The composition of the preserved larval shell of a migrant can therefore be used to identify its origin with an accuracy ranging

from 70 to 100% over 3,500 km in the Southwest Pacific Ocean. The most accurate classifications corresponded to geographically restricted vent fields, and required less predictors. This approach is justified if larval dispersal potential is known to be spatially restricted by complementary information (typically from larval dispersal models or population genetics), therefore providing an easier-to-perform and more precise determination of natal sites when the genetic data fails to provide robust assignments. Alternatively, determining the proportion of self-recruitment at a site of interest can be a means to estimate

the impact of habitat destruction and recovery (Mullineaux et al., 2010). This can be achieved for sites with high classification success rates, such as Kulo Lasi and Solwara 8. The next step is now to analyse the preserved pre-dispersal larval shell from juveniles after settlement (which corresponds to the encapsulated larval shells analysed here) to determine their origin from

our references. This method is promising and makes possible to examine the proportion of self-recruitment at any site, and assess the origin of migrants to study the connectivity and the ability for a population to recover after perturbations. This

information will be of utmost importance to understand the potential impacts of deep-sea mining ventures.

**Data availability**

All data are available in the main text, the supplementary materials and at https://doi.org/10.5281/zenodo.7828332.

**Author contribution**

VM, CC, MM, OR, DJ, TB, and TC contributed to the conceptualization of the study; VM, CP, FC, and TC designed the
methodology; VM, FC, YG, and OR participated in the investigation; VM performed the formal analysis and data curation; VM, DJ, TB, and TC completed the writing of the original draft, and VM, CP, FC, CC, MM, OR, DJ, TB, and TC reviewed and edited the manuscript.

**Acknowledgements**

This research was funded and carried out under the ANR CERBERUS project (ANR-17-CE02-0003). A CC-BY public
copyright license has been applied by the authors to the present document and will be applied to all subsequent versions up to the Author Accepted Manuscript arising from this submission, in accordance with the grant's open access conditions. Preprints are available at https://www.biorxiv.org/content/10.1101/2023.01.03.522618v1. We would like to thank the captains and crews of the French Research Vessel *L'Atalante* and the team in charge of the ROV Victor 6000 during the two parts of the Chubacarc cruise (Hourdez and Jollivet, 2019)https://doi.org/10.17600/18001111), without whom sampling would not have been
possible. We also wish to thank Nicolas Gayet for his help onboard with the PIF, and Camille Poitrimol for her help with collecting *I. nautilei*. Ship time and scientist travels were supported by the Flotte Océanique Française and the Centre National de la Recherche Scientifique (CNRS).

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

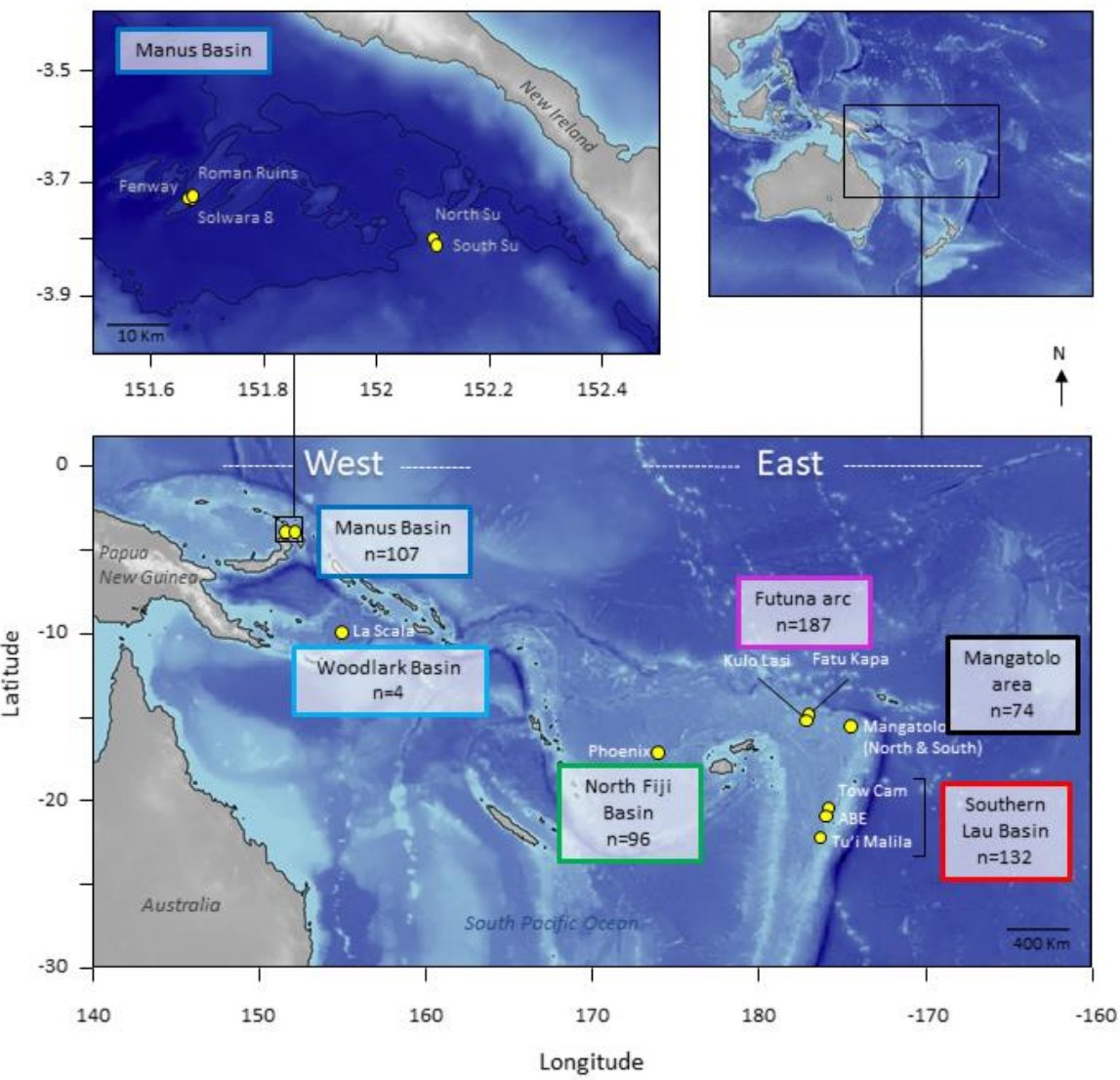

**Figure 1: Geographic map of the 14 sampling locations in the Southwest Pacific. The number of encapsulated larvae is indicated for each area. Although the newly discovered Mangatolo site is located in the northern part of the Lau Basin, we considered it as a separate area due to its greater distance with the other Lau sites. Maps generated using the marmap package for R (Pante and Simon-Bouhet, 2013) and bathymetry data from the NOAA ETOPO1 database (Amante and Eakins, 2009).**

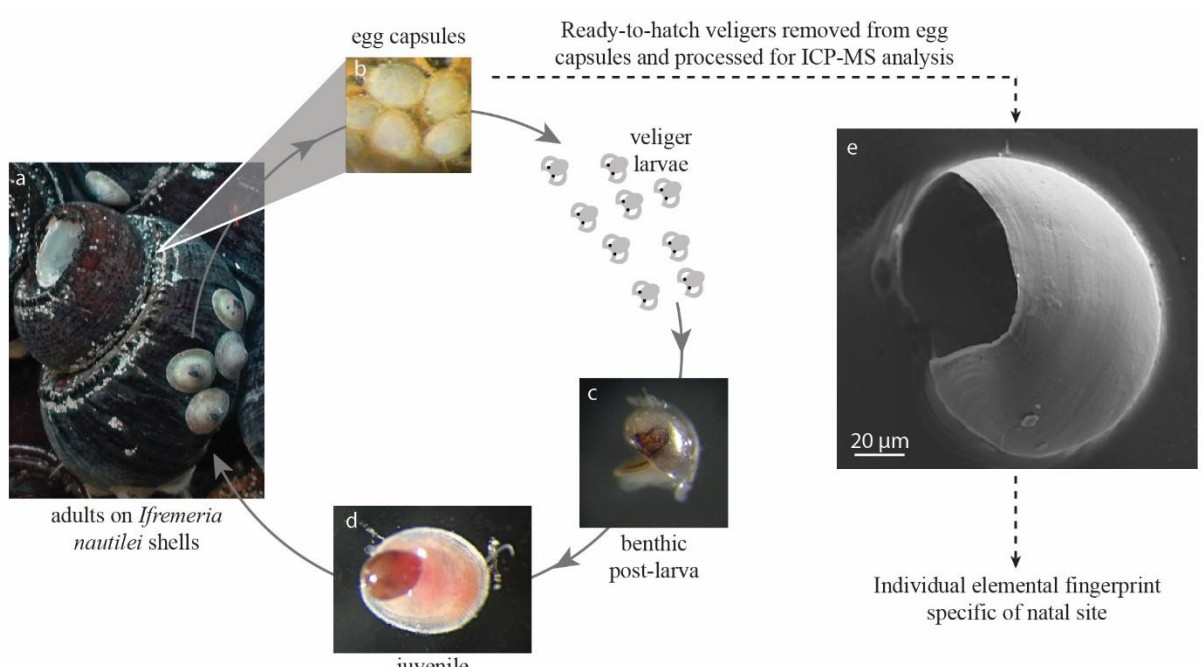

**Figure 2: Life cycle of *Shinkailepas tollmanni*. *Ifremeria nautilei* shell grooves (a) serve as a depository of *S. tollmanni* egg capsules (b) housing encapsulated larvae. When the capsules open, the shelled larvae disperse in seawater until metamorphosis when the individual reaches a hydrothermal site (settlement; c). The new recruit grows a juvenile shell while preserving its larval shell (d). A scanning electron microscope picture of an encapsulated larval shell (corresponding to the protoconch I; e) is presented (secondary electron mode, 15kV).**


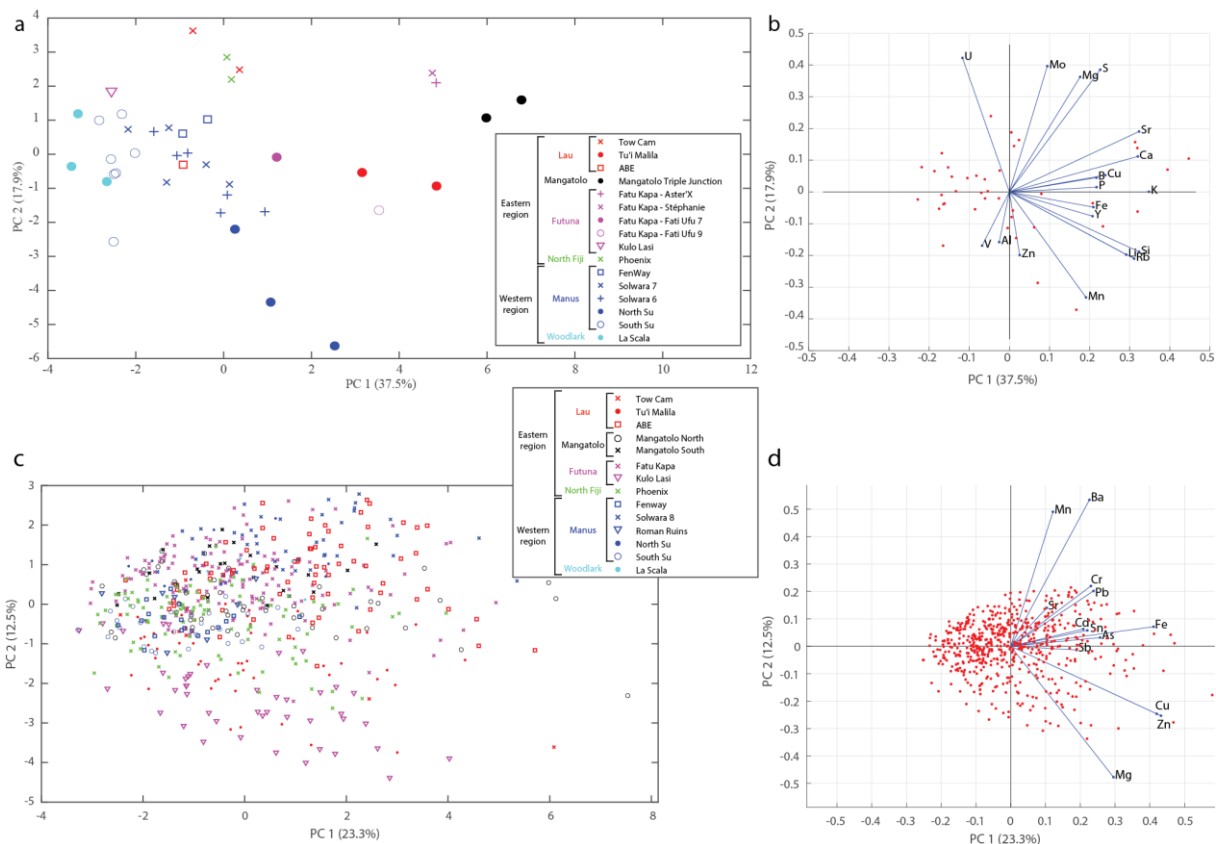

**Figure 3: Principal Component Analysis of *Ifremeria* habitat water compositions (a-b) and *Shinkailepas tollmanni* larval shell compositions (c-d) between sites, showing principal components 1 and 2. The legend gathers all sites by area (Lau, Mangatolo, Futuna, North Fiji, Manus and Woodlark) and by region (East and West). The variance explained by each principal component is given in parentheses (55.4% and 35.8% on the first two axes for habitat water and *S. tollmanni* larval shells, respectively).**

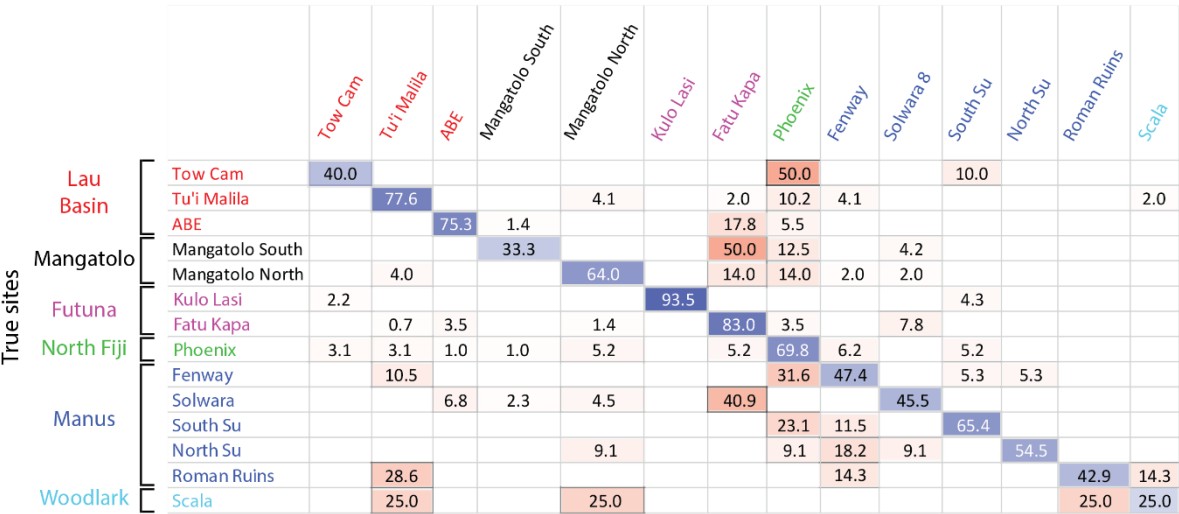

**Figure 4: Confusion matrix of the classification of *Shinkailepas tollmanni* encapsulated larvae to vent sites based on shell geochemistry. For each line (corresponding to one site), classification rates are indicated, with correct classification rates on the diagonal in blue and wrong classification rates outside the diagonal in red. The intensity of these colours is directly correlated to the classification rate (correct or wrong). This classification was made using Quadratic Support Vector Machines, reaching an overall 70.0% accuracy.**


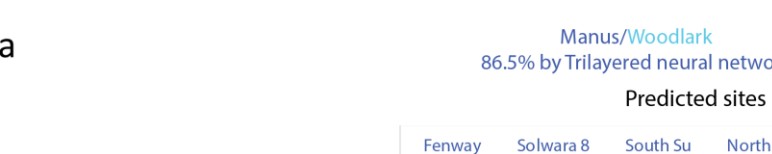

**Manus/Woodlark**
**86.5% by Trilayered neural network**

Predicted sites

| True sites | | Fenway | Solwara 8 | South Su | North Su | Roman Ruins | Scala |
|---|---|---|---|---|---|---|---|
| Manus | Fenway | 84.2 | | 10.5 | 5.3 | | |
| | Solwara 8 | | 100 | | | | |
| | South Su | 15.4 | | 84.6 | | | |
| | North Su | 18.2 | 18.2 | | 54.5 | | 9.1 |
| | Roman Ruins | | | | 14.3 | 71.4 | 14.3 |
| Woodlark | La Scala | | | | | 25 | 75.0 |

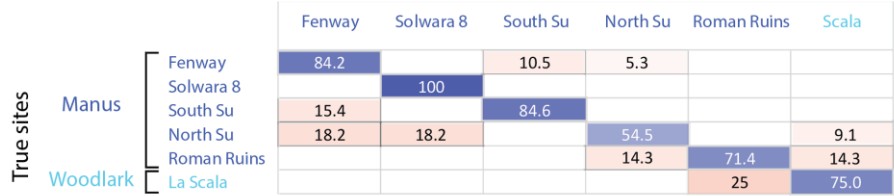

**Sites from East and Woodlark**
**77.1% by Quadratic SVM**

Predicted sites

| True sites | | Tow Cam | Tu'i Malila | ABE | Mangatolo South | Mangatolo North | Kulo Lasi | Fatu Kapa | Phoenix | Scala |
|---|---|---|---|---|---|---|---|---|---|---|
| Lau | Tow Cam | 10.0 | | 10.0 | | | | | 80.0 | |
| | Tu'i Malila | | 85.7 | | | 2.0 | | 2.0 | 10.2 | |
| | ABE | | | 71.2 | 1.4 | | | 24.7 | 2.7 | |
| Mangatolo | Mangatolo South | | | | 29.2 | | | 50.0 | 20.8 | |
| | Mangatolo North | | 4.0 | 4.0 | | 62.0 | | 12.0 | 18.0 | |
| Futuna | Kulo Lasi | 2.2 | | | | | 95.7 | | 2.2 | |
| | Fatu Kapa | | 0.7 | 4.3 | 0.7 | 1.4 | 12.0 | 86.5 | 6.4 | |
| North Fiji | Phoenix | 1.0 | 2.1 | 2.1 | 1.0 | 7.3 | | 4.2 | 82.3 | |
| Woodlark | La Scala | | 50.0 | | | | | | | 50.0 |

**Regions from East and Woodlark**
**75.3% by Quadratic SVM**

Predicted areas

| True areas | Lau | Mangatolo | Futuna | North Fiji | Woodlark |
|---|---|---|---|---|---|
| Lau | 68.2 | 6.1 | 14.4 | 11.4 | |
| Mangatolo | 6.8 | 51.4 | 21.6 | 20.3 | |
| Futuna | 4.3 | 4.3 | 87.7 | 3.7 | |
| North Fiji | 3.1 | 9.4 | 7.3 | 80.2 | |
| Woodlark | 25.0 | 25.0 | | | 50.0 |

**Figure 5: Confusion matrices of the classification of *Shinkailepas tollmanni* encapsulated larvae based on shell geochemistry at the region scale. a: Using Mn and Ba as predictors for sites sampled in the Manus and Woodlark areas only (western region). b: Using Mg, Mn, Fe, Zn, Sr, Ba and Pb as predictors for sites from Woodlark, North Fiji, Futuna, Lau, and Mangatolo areas (eastern region and Woodlark). c: Using Mg, Mn, Fe, Zn, Sr, Ba and Pb as predictors for all areas except Manus (eastern region and Woodlark). For each line, corresponding to one site (a, c) or one area (b), classification rates are indicated, with correct classification rates on the diagonal in blue and wrong classification rates outside the diagonal in red. The intensity of these colours is directly correlated to the classification rate (correct or wrong).**

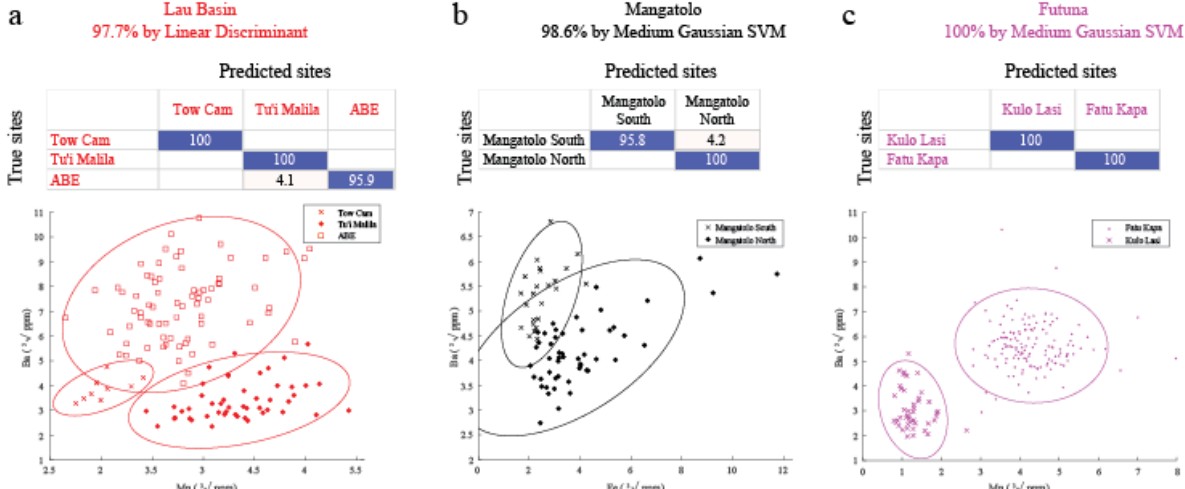

**Figure 6: Confusion matrices of the classification of *Shinkailepas tollmanni* encapsulated larvae based on shell geochemistry between sites for each eastern area and scatter plots of corresponding datasets (a: Lau; b: Mangatolo; c: Futuna volcanic arc). For each line of the matrices (corresponding to one site), percentage of correct classification are indicated in blue and wrong classification in red. For each matrix, the precision and type of model are indicated. The elements kept to build these models are Mn and Ba for Lau Basin and Futuna volcanic arc, and Fe and Ba for Mangatolo, as illustrated by the scatter plots.**


**Table 1:** Correlations of elemental ratios to Ca between habitat water and larval shells.

| Elemental ratios | $r^2$ Pearson | *p-value* Pearson | *rho* Spearman | *p-value* Spearman |
|---|---|---|---|---|
| Mg/Ca | 0.002 | 0.3 | 0.015 | 0.731 |
| Mn/Ca | 0.001 | 0.47 | -0.072 | 0.096 |
| Fe/Ca | 0.145 | < 0.001 | 0.580 | < 0.001 |
| Cu/Ca | 0.022 | < 0.001 | 0.012 | 0.786 |
| Zn/Ca | 0.001 | 0.42 | 0.030 | 0.481 |
| Sr/Ca | 0.001 | 0.57 | 0.044 | 0.312 |

**Table 2:** Correlations of elemental ratios to Mn between habitat water and larval shells.

| Elemental ratios | $r^2$ Pearson | p-value Pearson | rho Spearman | p-value Spearman |
|---|---|---|---|---|
| Mg/Mn | 0.492 | < 0.001 | 0.449 | < 0.001 |
| Fe/Mn | 0.004 | 0.124 | -0.154 | < 0.001 |
| Cu/Mn | 0.429 | < 0.001 | 0.305 | < 0.001 |
| Zn/Mn | 0.448 | < 0.001 | 0.256 | < 0.001 |
| Sr/Mn | 0.641 | < 0.001 | 0.639 | < 0.001 |