# Peer review of "A step towards measuring connectivity in the deep sea: elemental fingerprints of mollusk larval shells discriminate hydrothermal vent sites"

_EGUsphere, 2023_

## Author Comment (AC1)

**a**

| True areas | Predicted areas | | | | |
|---|---|---|---|---|---|
| | Futuna | Lau Basin | Mangatolo | North Fiji | Woodlark |
| Futuna | 80 | | | 20 | |
| Lau Basin | 40 | 60 | | | |
| Mangatolo | | 50 | 50 | | |
| North Fiji | | | | 100 | |
| Woodlark | 33.3 | | | | 66.7 |

**b**

| True areas | Predicted areas | | | | |
|---|---|---|---|---|---|
| | Futuna | Lau Basin | Mangatolo | North Fiji | Woodlark |
| Futuna | 4 | | | 1 | |
| Lau Basin | 2 | 3 | | | |
| Mangatolo | | 1 | 1 | | |
| North Fiji | | | | 2 | |
| Woodlark | 1 | | | | 2 |

**c**

| True areas | Predicted areas | |
|---|---|---|
| | Manus | Woodlark |
| Manus | 91.7 | 8.3 |
| Woodlark | 33.3 | 66.7 |

**d**

| True areas | Predicted areas | |
|---|---|---|
| | Manus | Woodlark |
| Manus | 11 | 1 |
| Woodlark | 1 | 2 |

**Figure 1:** Confusion matrices of the classification of habitat water measurements using Mg, Mn, Fe, Cu, Zn, and Sr for Eastern areas (a-b) and Western areas (c-d). For each line, corresponding to one area, classification rates (a, c) and number of measures (b, d) are indicated, with correct classification on the diagonal in blue and wrong classification outside the diagonal in red.

---

## Author Response (AR1)

Revisions made from reviewer comments

**RC1**

**General comments**

This work represents an important first step in the application of trace-elemental fingerprinting to deep-sea hydrothermal vent connectivity. The authors have selected a suitable test species and studied an impressive array of sites (14). I think this effort is a precursor to many important applications.

Although the authors present deep seabed mining as the key motivator , I would argue that there are many additional reasons for wanting to know about connectivity of vents – from basic science, to biodiversity conservation (30x30 goals), to addressing consequences of climate change on connectivity (e.g., Mitarai's work in Levin et al. 2020 DOI: 10.1111/gcb.15223), vent roles in the carbon cycle etc. I would encourage at least mention of these other motivators. Within the mining realm, in addition to understanding consequences of mining disturbance, the authors should point out the importance of connectivity data to the designation of no-mining protected areas (APEIs) and also reference zone PRZ and IRZ (preservation and impact) designation.

**We added some references to include these in the second paragraph of the introduction (L. 39-45). Also, we mention protected areas when we give details on how we can use elemental fingerprinting (see last general comment below).**

Please consider adding to the end of the introduction a paragraph that clearly lays out the goals or objectives of the research. This could be in the form of questions, hypotheses or other… but should frame the science around the data presented in the paper. E.g., does the water chemistry of 'habitat water' differ among vent sites where *S. tollmanni* egg capsules occur? What elements are key to distinguishing sites? Does the trace element signatures of *S. tollmanni* larval shell reflect the habitat water chemistry? Are there specific sites or scales of connectivity where the application of trace elemental fingerprinting to the vent systems is likely to most reliable? What elements are key to this distinction? Some of these questions show up as the headings in the results section – but should be presented earlier.

**We added a paragraph at the end of the introduction (L.115-120) to list the main questions the manuscript aims at answering: "To explore the use of elemental fingerprints of larval shells in this species, several questions were investigated. Are there enough chemical contrasts between hydrothermal vent sites for elemental fingerprinting? Do elemental fingerprints of larval shells reflect their natal place? What are the main elements responsible for the elemental fingerprint of larval shells? At which spatial scale is elemental fingerprint more accurate? Does the elemental fingerprint of larval shells correspond to that of habitat water, in the sense that the habitat water fingerprint could be used as a reference for the shell fingerprint?".**

Tell us a little bit more about the study species *Shinkailepas tollmanni* – its distribution, and its host *Iffremeria* distribution. Is the relationship obligate? Is anything known about depth ranges, longevity, development time, feeding mode, planktonic duration? How does its life

history affect inferences about connectivity? Will information about mollusc/gastropod connectivity be relevant to other vent taxa – what will or won't?

**We added available information on the biology of the studied species as requested by RC1 in a new first sub-section of the Materials and Methods (section 2.1, L.122-156). Most questions asked by RC1 about its life history traits were addressed, except the question of its longevity, for which no report exists, to our knowledge.**

**Unfortunately, for now, we cannot reliably give information on how connectivity interpretation inferred from elemental fingerprinting assignment (we don't have the necessary data yet; see reply below) would be 'relevant to other vent taxa'. In particular, we don't have the data to discuss the possibility of using elemental fingerprint references from *S. tollmanni* to assign the origin of individuals from other taxa occurring on the same vent sites.**

Consider discussion which of the study sites are targeted for mining, and which might serve as source populations. IUCN has red listed some species like the scaly foot snail based on their limited occurrence primarily in areas targeted for mining. I realize you don't have any source or sink data generated yet but it might be useful to explain how this precursor work can lead to analyses that inform identification of vulnerable species.

**The identification of source and/or sink populations is currently impossible, as we have no assignment of juvenile individuals from our method. We however added in the introduction specifics to how the assignment of juveniles from elemental fingerprints of their larval shells can indeed identify these populations, and help determine which of these sites should be particularly critical for the survival of the studied species (L.99-106).**

**Specific comments.**

Title – please only hyphenate deep sea when it is used as a double adjective. Here it is a noun and should not be hyphenated.

**Changes made.**

Abstract: Line 19. The presence of capsules not only facilitates sampling, it means the embryos develop in a fixed location – and form carbonates whose trace elemental signature could reflect that location.

**Changes made.**

Line 30 – Even vent sites on mid ocean ridges far from coastlines are of interest to commercial mining (ISA contracts on Mid Atlantic Ridge and Southwest Indian Ridge).

**We removed the precision of 'close to coastlines'.**

Line 65 – replace 'minute' with a more accurate indication of size…. 100 micron?

**Changes made (now L.99).**

Line 72 – replace 'which carbonate shells 'with 'whose carbonate shells'

**Changes made (now L.112).**

Line 263  Simmonds et al. should come out of the parentheses.

**Changes made (now L.369).**

Note that Levin 2006  (doi:10.1093/icb/icj024) in discussing future directions for larval dispersal studies in a larval dispersal review  wrote *How much larval exchange occurs within and among reducing ecosystems such as vents, seeps, and whale falls? Analysis of short-term larval exchange among seep or hydrothermal vent ecosystemsmight be tractable if these impart distinct trace element signatures to larval shells.*

I think this paper is really the first to tackle this problem.  (17 years later).

**This is a nice show of interest to our work, as well as a long overdue answer to this suggestion! We added this reference (L.64-65).**

Fig. 4 and 5 caption.  Indicate what the lighter and darker shades of blue and red mean.

**Changes made.**

The supplemental information needs a table of contents as it is extensive.

**Changes made.**

Table S4 should replace the ',' with decimal points.

**Changes made. The table has been included in the main manuscript as Table 1 (see next comment).**

The comparison of habitat water to larval shell chemistry should be included in the main paper.

**We now included the precisions of the comparison of habitat water and larval shell chemistry in the main manuscript (L.379-389), along with Tables 1 and 2.**

Overall this is a significant contribution in need of relatively minor revision.

 **We thank you for your interest.**

**RC2**

This is an interesting study exploring the use of the elemental fingerprint of gastropod larval shells to predict their origin at certain hydrothermal vent sites. Overall, the study is well executed and described, and is worth publication.

**We thank you for your interest.**

There is, however, one issue that the authors should check. In the last paragraph before the conclusions (lines 292-293), they write '*our data clearly evidenced that chemical composition of habitat water is not a reliable predictor of shell chemistry, and thus cannot be used to assign the origin of individuals*'. This raises the question of the overall role of water chemistry in larval shell chemistry, which in turn puts the whole approach into question: if larval shell chemistry is largely biologically controlled, it would have little geographic meaning.

What the authors could do is the following test: do the same set of tests for habitat water chemistry as they have done for larval shell chemistry, and check if the geographic structure found through both sets of tests match. It is entirely possible (and in my view likely) that water chemistry does indeed determine larval shell chemistry. However, there could be various, possibly biologically controlled, inter-dependencies among the elements that make it difficult to find a straightforward, one-to-one match between water and shell chemistry. By comparing the geographic structure of habitat water chemistry vs. larval shell chemistry, that problem could be avoided.

**As mentioned in the open discussion phase (AC1), we have now clarified that 1) habitat water composition does differ between hydrothermal sites, 2) it does influence the composition of larval shells, but 3) shell chemistry is also controlled by biological processes, meaning that habitat water alone, although largely responsible for differences in shell composition, is not sufficient to be used directly to assign the origin of larvae. Because of the biological processes that alter the link between water and shell chemistry, site-specific references must be built using the local composition of shells, not water. We have now clarified this point throughout (L.118-120, 418-420, 422-423).**

**We have also performed the analysis suggested in this comment (see AC1), but as there is too little power due to the lower number of measurements (habitat water was analyzed from a single sample per site, compared to shell chemistry analyses), we have chosen not to add it to the manuscript (as stated L.284-285).**

---

## Author Response (AR2)

Corrections requested from the editor

Thank you for submitting your interesting and pioneering paper to Biogeosciences. I have read the revision, and I am happy to inform you that your paper is now accepted for publication pending one minor technical correction in line 215: I guess a superscript is missing.

**Indeed, this typo has been modified, thank you. Now the correct figure is 10.2 $10^3$.**